# ShinyLUTS—A Shiny web application for structured data management and analysis for patients with lower urinary tract symptoms (LUTS)

**Christoph-Alexander Joachim von Klot**[1]*, **Cornelius Köpp**[1,2], **Markus Antonius Kuczyk**[1], **Mathias Wolters**[1]

1 Clinic for Urology and Urological Oncology, Medical School Hanover, Hanover, Lower Saxony, Germany,
2 FHDW University of Applied Sciences in Hanover, Hanover, Lower Saxony, Germany

* klot.christoph@mh-hannover.de

**Data Availability Statement:** All relevant data are within the manuscript and its Supporting Information files. The manuscript is about a database tool the code is provided in in

## Abstract

### Objectives

Clinical, time-dependent, therapeutic and diagnostic data of patients with LUTS are highly complex. To better manage these data for therapists' and researchers' we developed the application ShinyLUTS.

### Material and methods

The statistical programming language R and the framework Shiny were used to develop a platform for data entry, monitoring of therapy and scientific data analysis. As part of a use case, ShinyLUTS was evaluated for patients with non-neurogenic LUTS who were receiving Rezum™ therapy.

### Results

The final database on patients with LUTS comprised a total of 8.118 time-dependent parameters in 11 data tables. Data entry, monitoring of therapy as well as data retrieval for scientific use, was deemed feasible, intuitive and well accepted.

### Conclusion

The ShinyLUTs application presented here is suitable for collecting, archiving, and managing complex data on patients with LUTS. Aside from the implementation in a scientific workflow, it is suited for monitoring treatment of patients and functional results over time.

supplementary file (s1_file). Clinical data/patient data itself is not published in this manuscript and can therefore not be provided. The current tool uses 'live' uncensored/non-anonymized patient data which cannot be provided for ethical/data protection reasons. Patient data is not published in this article but it was published in a previous paper (Winkler T, Klot CAJ von, Madersbacher S, Kuczyk MA, Wolters M. Rezum water vapor thermal therapy for treatment of lower urinary tract symptoms: A retrospective single-centre analysis from a german high-volume centre. Lee RK, editor. PLOS ONE [Internet]. 2023 Jan;18(1):e0279883. Available from: https://doi.org/10.1371/journal.pone.0279883).

**Funding:** The author(s) received no specific funding for this work.

**Competing interests:** The authors have declared that no competing interests exist.

## Introduction

Clinical data of patients with lower urinary tract symptoms (LUTS) is highly complex if all aspects are to be taken into account, such as patients' medication, urodynamic diagnostics, questionnaires, ultrasound results, laboratory results, micturition diary entries and many more. Additionally, a plethora of surgical techniques including transurethral resection, enucleation, aquaablation or Rezum™ ablation can be applied, each of them with its own set of parameters. The fact that each diagnostic or therapeutic data point has a temporal dimension, can be repeated or can change over time, makes structured data collection and evaluation considerably difficult. Urodynamic measurements generate vast amounts of data, are sometimes repeated and have a specific temporal relationship with therapeutic events or clinical outcomes, such as questionnaires. For clinicians it is sometimes difficult to maintain a reliable overview. Spread sheet solutions like Microsoft® Excel, Google Sheets or LibreOffice Calc are widely available. However, branched and high-dimensional data of this type cannot be adequately captured in a two-dimensional matrix due to inevitable redundancies and conflicts with normalization rules [1]. A number of commercial or non-commercial relational and non-relational database solutions are available. A comparably high level of prior knowledge in the field of data processing, database management systems (DBMS) and sometimes even application programming interfaces (API) is required for familiarization with this topic [2, 3]. Some commercial electronic health record systems and systems like REDCap® need to be setup, need to be integrated into the institutional IT infrastructure, may come with costly license fees and are not tailored specifically to patients with LUTS. In addition, an existing database may need constant maintenance, especially to give the researcher the possibility to adapt and change the database structure at any time [4]. A scientific database of this kind is generally not suitable for monitoring the patient's therapy.

R is one of the most widely used statistical programming languages among researchers. Shiny is an R package that provides a framework for creating web-based interactive applications. A number of scientific papers on the clinical application of Shiny have been published recently [5–7]. The vast options of this package have not yet found their way into the field of urology and, in particular, not into the treatment or data collection of patients with LUTS.

We formulated the following functional and technical requirements for our database solution: The new tool should be usable for scientific research as well as for every day practice during the treatment of patients with LUTS. Data should be centralized and stored in accordance with current data protection regulations. A clear summary of the data should be shown during the entry of new data for the respective patient. If possible, a graphical representation of numerical data should be implemented. The data input should be controllable with regard to its plausibility. The input of redundant data should be avoided. The database structure should be freely customizable by the user, especially the addition of new columns or entire data tables should be possible. Cooperative work of multiple physicians and researchers should be possible for both evaluation and data entry. Direct access and manipulation of the data via commonly used spread sheet tools (such as Microsoft® Excel) should be provided. The usability of the new database tool is tested by means of integration in the workflow of patient interaction and therapeutic decisions and also as part of a scientific project on patients with LUTS. Finally, all the above-mentioned functionality should be usable without any prior technical knowledge on database systems.

Here we present the development and testing of a relational database model that enables researchers to enter highly complex and multidimensional data for patients with LUTS via a greatly simplified web interface into a modifiable database that can be used for daily clinical practice as well as large scale scientific evaluation of data. For this we use the Shiny framework of the statistical programming language R [8].

## Material and methods

### Patients

The acquisition of data for the use case of a scientific project was performed using a mono-centric approach from patient records, as well as with data derived from urodynamic measurements. Only patients with non-neurogenic male LUTS, including benign prostatic obstruction (BPO), were included in the first dataset. The assessment of clinical data was not subject to the current research work on the ShinyLUTS data tool. Ethical approval and patient consent were obtained from the publication of clinical data and clinical results by Winkler et al. [9].

### User interface

The R-Shiny framework was used for the creation of the web based general user interface (GUI) [8]. The applicability of the tool was independently evaluated by 3 physicians (CAJK, TW [9], MW).

### Database backend

Data storage was tested in the format of comma separated values (.csv), SQLite (SQLite Release 3.38.0) [10], or Microsoft® Excel workbook with multiple spreadsheets (.xlsx).

### Data protection

Data was stored in accordance with the current European General Data Protection Regulation (GDPR, Regulation 2016/679).

### Technical aspects

For the creation of the web-based GUI, the packages: Shiny [8], Shinymanager [11] and DT [12] and for the connection to SQLite and Microsoft® Excel worksheet files, the packages RSQLite [13] and openxlsx [14] were used.

We applied filtering, grouping and aggregating functions from the the r-dplyr [15] package prior to data analysis. For a live graphical representation of the data within the GUI, the R-package ggplot2 was used [16]. RStudio version 2021.9.0.351 [17] and R version 4.1.1 [18] were used as integrated development environment.

## Results

### Dimension of the final database

At the time of the first publication, the database consisted of 7 tables with 8,118 data points for 94 patients, 92 of which were to be evaluated in the publication [9]. As of today, the growing database organized by the ShinyLUTS tool includes 13 tables with 65,806 data points for a total of 243 patients. The pre-specified technical requirement of a growing database structure was fulfilled.

### General user interface (GUI)

The final interface shows the patient base table with mandatory fields for the hospital-specific ID and date of birth. In a separate list, an aggregated overview of all clinical data tables and the date of the last entry in each table are shown. Each clinical data table can be selected individually for data manipulation. Lastly, a graphical representation of numerical data over time was implemented. Fig 1 shows the ShinyLUTS web interface in its final application form with a detailed description of functionality.

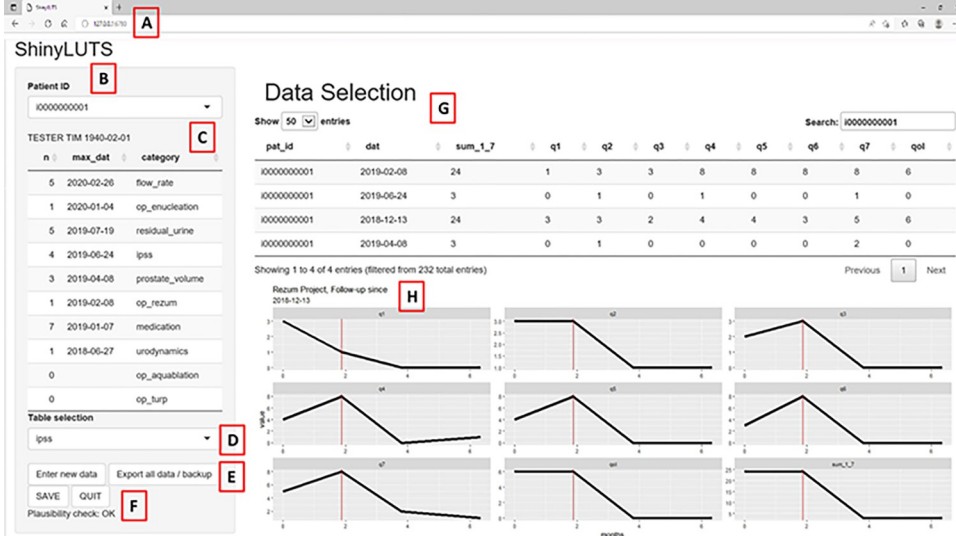

**Fig 1. General User Interface (GUI).** The Figure shows the ShinyLUTS application interface. A) Browser window. B) Patient ID as a dropdown menu that allows manual input with autocompletion. C) A Summary of all data tables for the selected patient, defined by the patient ID in the dropdown menu (B): The summary shows the number of entries for each data table (n), the name of the data table (category), and the date of the last entry for each table (max_dat). This allowed for a quick overview of each patient. This aggregated view allows the user to quickly see how much data are available for the selected patient and when the last entry is dated. D) Dropdown menu for selecting the data table to be viewed or edited (Data Selection). E) Action buttons have various functions. 'Enter new data' will add a new empty row to the data table pre-selected in dropdown menu (D) below the data table (G), starting with the patient ID pre-selected in dropdown menu (B). Patient ID was not prefilled when the patient base table was selected. 'Export all data / backup' will create a dated backup folder with a copy of the current database. The 'SAVE' and 'QUIT' buttons have corresponding functions. (F) Information box displaying a plausibility check. Because the underlying database is based on an Excel worksheet with the claim to be as free as possible, the usual schema with restrictions on data consistency, normalization rules, or data types are not possible at the database level. Currently, our tool is limited to checking the patient ID of the patient base table and clinical data table as part of the plausibility check. G) Selected data. The selected patient ID (B) and selected data table (D) determine the content of this table. In this view, data can be manipulated and new rows can be inserted via the 'Enter new data' action button. H) Graphical representation of data over time in relation to specific therapeutic interventions (Rezum™ in this case). The graphs show all numerical values of the selected data table with time/date references for the selected patient. This graphical representation was designed to monitor patient data over time in relation to a specific intervention.

## Data structure

In order for the automated, flexible implementation of the database structure to work, several rules were found to be useful for implementation: 1) Column or table names were not allowed to have special or white-space characters. This rule was especially important to ensure a seamless integration into the R environment. 2) All tables must have at least a patient id and a date column with the exception of the patient base table. The id column and the date column must have the same name in all tables. This was essential to make relational queries possible and to check for plausibility between the patient base table and the clinical data tables. We adhered to strict column name conventions for the patients' ID and the date column to avoid the definition of aliases. 3) The database consists of exactly one patient base table and one or several clinical data tables. Additional tables are allowed for documentation and experimentation purposes. They will however not be displayed in the Shiny web GUI if the 'id' or 'date' column is missing. Additional tables or columns that do not adhere to the pre-specified nomenclature are automatically excluded from the GUI of the web-application, but will be preserved in the backend database. This architecture gave the researcher the opportunity to freely decide which data tables are to be included or excluded in the GUI interface. Also, researchers were able to

use the full set of Microsoft® Excel functionality (including calculation, formatting and plotting of data) without interfering with the original data in the data tables. The predefined requirement to ensure spreadsheet functionality was thus fully met.

## Creation of tables and data entries

The first model for data management was a server based relational SQL (Structured query language) database. In a completely different approach, we tested Microsoft® data tables in.xlsx- and.csv formatting. While SQL databases give great functionality with regard to data integrity, plausibility, the avoidance of redundant or inconsistent data and also facilitates normalization rules and constraints, we changed the data base backend to a simple spread sheet matrix located on a network drive which was maintained mainly by Microsoft® Excel. By doing so, clinicians and researchers were able to freely create tables, new fields and columns as desired. All tables and fields were automatically integrated into the ShinyLUTS application as long as they fulfilled the above-mentioned formatting rules. Following this approach, the restraints of an SQL database could no longer be implemented. However, we created an adaptable plausibility check on the level of the application as info box inside the ShinyLUTS GUI. At the current development state, the application will check (but not prevent) for inconsistencies between patient ids in the patient base table and in the clinical data table.

Particularly noteworthy was the fact that in addition to the core task of data collection, the application was also perceived as very suitable for monitoring the patient's therapy. According to the physicians, patient's summary and the graphical representation of the data, which was specifically incorporated into the application for this purpose, provided a clear advantage for the treatment of patients in everyday clinical practice. The GUI structure and data entry was perceived as intuitive and easy to manage. I.e., after implementation of the web based interface, urologic researchers were able to maintain, fill and alter the database structure without further technical assistance as was a pre-specified prerequisite of this project.

## Data protection

Data protection was facilitated by the fact that server-based data entry was not performed and instead the data was kept as tables on a network drive. This allowed the data protection measures inherent in the institution to take effect. Access was restricted to the user without a limited number of users. Thus, no new privacy concerns arose from the use of ShinyLUTS. Data protection regulations vary from country to country. In this case, the European Union's data protection requirements applied, which in Europe is the General Data Protection Regulation (GDPR, Regulation 2016/679, https://www.consilium.europa.eu/en/policies/data-protection/data-protection-regulation/).

## Use case

As part of an initial retrospective research project, we integrated the ShinyLUTS tool, not only into the workflow of every day treatment of patients but also into a workflow with a Rmarkdown script for research purposes (RMarkdown is a markup language that allows to combine scientific manuscript writing and statistical calculations with the R programming language in one document [19]). Data were selected, filtered and aggregated according to the research question. The researchers were able to observe the progress of the data acquisition and changes in the statistical analyses, figures and tables directly and dynamically over time. Fig 2 shows a schematic of the data flow as well as the corresponding prior knowledge required in each case, starting with the input of the raw data and ending with the evaluation of the data in the context of a Rmarkdown design. In total, the streamlined workflow, leading directly from raw data

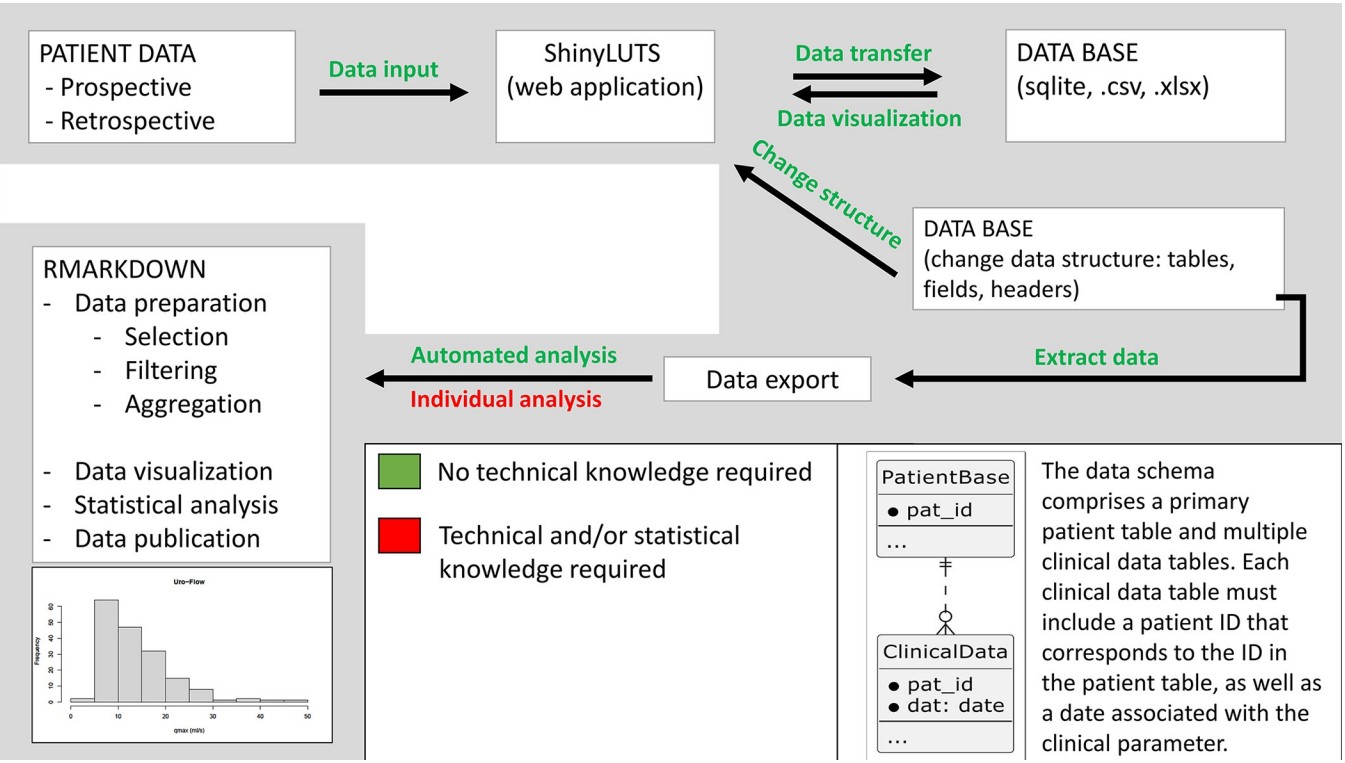

**Fig 2. Data flow schematic.** Workflow for data processing and evaluation involving ShinyLUTS. Substeps that require special prior knowledge in the area of data analysis or database management (red) are distinguished from substeps that can be performed by a medical professional without any prior knowledge (green). The input of raw data is done in the ShinyLUTS web application (parallel input in known spreadsheet solutions like Microsoft® Excel is possible). The transfer of the data into the respective data table of the database is done automatically. It is also possible to change the data structure completely, insert new fields or add data tables. ShinyLUTS integrates all changes automatically into the web interface. Our workflow provides two potential ways of processing the raw data in the database: First, an automated aggregation and summary of the data, both in text and tabular form, as well as graphically. This includes summaries, statistics on distributions, totals, means, standard deviations, etc. I.e. an automated descriptive summary of all numerical data contained in the database. This evaluation is automated and does not require any prior technical knowledge. The second form of evaluation, is the analysis with regard to the specific question to the data set. Individual statistical calculation of data, selected and filtered according to certain rules, still requires knowledge of database querying and statistics and cannot be automated.

entry to statistical and graphical results was perceived helpful and was readily integrated into the final research paper which was published by Winkler et al. [9].

## Discussion

The initial idea of the need for a database tool for managing the data of patients with LUTS arose in the clinical context of patient treatment, as well as during the planning of several research projects. The comparatively complex data from the urodynamic examinations alone, combined with the data on medication, therapeutic interventions and changes in the functional data over time, often led clinicians and scientists to the limits of their abilities.

Our major goal of the ShinyLUTS data tool was to create an environment, where urologists were able to create and manage complex data structures as well as insert new data without any prior knowledge on programming, database design (and maintenance) and information technology in general. To our knowledge, an application of this type does not yet exist for patients with LUTS or for the documentation of functional data in urology in general.

Our experience with the here outlined project showed that both goals, i.e., the improvement of the scientific data flow as well as the improved clinical monitoring of patients with LUTS, have been achieved.

Probably one of the major advantages of our application in scientific work was the possibility of centralized data entry by all participating physicians. The application could be used without having to pay attention to versions of different data tables. Data entry as well as the modification of data or even the extension of the entire data structure was possible without further knowledge of database management. This also applied to the extraction of the data and even to a ready-made analysis for the visualization of standard queries such as sum ratios, frequencies and density distributions.

One of the biggest challenges of this project was to combine the ideas of a medical scientist with the possibilities of modern database systems, which are almost exclusively understood and used by technical specialists. Physicians and researchers need the opportunity to work in understandable tables. A direct view of the data is just one of the benefits researchers demand. Changing the data, copying and pasting large blocks of data or entering new or external tables by hand is comparatively easy in Microsoft® Excel but within the setting of a server-based relational database these tasks would only be possible with special technical expertise. On the other hand, a relational, server-based database is necessary for managing highly complex and large-scale data.

In this project we believe we have combined the best of these two worlds. I.e., the functionality and simplicity of a Microsoft® Excel environment with the possibilities to organize very large amounts of data in numerous tables and to link them relationally without increasing the complexity. Being able to enter highly complex clinical data for patients with LUTS, while keeping the view on the data simple and manageable even when the amount of data greatly increases, was finally made possible and solved a variety of problems faced by many physicians and researchers in the past.

The the current version of ShinyLUTS enables clinicians to implement the tool without a special IT or server infrastructure and without any setup. The data structure is defined by Excel spreadsheet data and later data entry is maintained via the ShinyLUTS tool. It is even possible to add full data sets that were collected outside the ShinyLUTS tool. This means that, for example, laboratory data or data on patients' medication may have been collected as part of a clinical research within a statistical application environment or any spreadsheet application. Copying these data into a new data table within the xlsx.-file for the ShinyLUTS application would immediately result in an integration of the data, the possibility to maintain and add to the data via ShinyLUTS and also to see the newly added data in relation to preexisting data on treatment modalities (as long as the added data includes a corresponding 'id' and 'date' column.

A few limitations have to be mentioned: To our knowledge, the presented workflow with its central element, the web application 'ShinyLUTS', is suitable for any collection of functional and therapeutic data on patients with LUTS. However, we cannot completely exclude the possibility that special questions may require an even more in-depth or branched data structure, which we have not yet mapped here in this application. As far as we know, our application can be used, without prior adaptation, for all issues related to patients with LUTS.

The evaluation of the here presented application started with a SQL database server as backend which allowed almost unlimited simultaneous data entry. The requirement of the physicians to be able to intervene in the database structure as well as to be allowed to carry out direct manipulations via spread-sheet applications, finally led to storage of the data in.csv or.xlsx format, which can guarantee the cooperative, but not the simultaneous manipulation of the data in every case. Therefore, the option of simultaneous multi center data collection was also omitted. In the same way, the control of data integrity, relations between primary and external conclusions, and the plausibility of the data through the user's ability to access the data must be viewed critically. Our solution does not prevent inconsistencies in the data, but points them out.

Another limitation of functionality is the need for an often individualized analysis of the data, prior to publication of results which still requires prior technical knowledge to some extend. First, urologists must be able to select data from the database to answer the predefined questions. For this purpose, data points may have to be checked for interdependencies. A frequent problem here is the dependence of parameters on defined points in time. For example: for the evaluation of functional values before and after therapy for LUTS, the parameters had to be set in relation to the respective date of intervention. Furthermore, rules have to be applied that define time periods in which a value counts as preoperative or postoperative. A data query of this kind requires an idea of the scientific question to be answered, as well as a technical knowledge of filtering, grouping or aggregation functions or the handling of PIVOT tables. This type of individual data retrieval and evaluation as well as the fitting of statistical models to a specific scientific question cannot be automated.

We believe that our tool is worth implementing for various reasons: We have observed a significant improvement in the management of clinical data. In particular, patient monitoring has improved. For the first time, doctors have the opportunity to view all clinical data, such as therapeutic response, measurement or laboratory values when advising patients. We gained better work experience and increased clinical data availability. Our solution is scalable without increasing the complexity used by researchers and clinicians in the past, and was successful in the first scientific use case.

## Conclusions

The ShinyLUTs application presented here is suitable for collecting, archiving, and managing the full range of data collection on patients with LUTS. Aside from scientific work, it is also suited for monitoring patient treatment. The application does not require any technical knowledge, needs no setup and is therefore well suited for cooperative medical research and monitoring of patients' data without prior technical knowledge.

The valuable insight gained here in the course of the project about the requirements of physicians and researchers in the collection and analysis of clinical data is most likely not limited to patients with non-neurogenic male LUTS. As an outlook, we aim at a generalized form of our database solution for a variety of clinical projects.

## Supporting information

**S1 File. Raw code / ShinyLUTS.** The file contains the current version of raw code for the ShinyLUTS application. The current version is subjected to constant change as the tool is further developed.
(TXT)

## Author Contributions

**Conceptualization:** Christoph-Alexander Joachim von Klot.

**Formal analysis:** Christoph-Alexander Joachim von Klot, Cornelius Köpp.

**Methodology:** Christoph-Alexander Joachim von Klot, Cornelius Köpp.

**Project administration:** Christoph-Alexander Joachim von Klot.

**Resources:** Markus Antonius Kuczyk, Mathias Wolters.

**Software:** Christoph-Alexander Joachim von Klot.

**Supervision:** Cornelius Köpp, Markus Antonius Kuczyk, Mathias Wolters.

**Validation:** Christoph-Alexander Joachim von Klot, Cornelius Köpp, Markus Antonius Kuczyk, Mathias Wolters.

**Visualization:** Christoph-Alexander Joachim von Klot.

**Writing – original draft:** Christoph-Alexander Joachim von Klot.

**Writing – review & editing:** Christoph-Alexander Joachim von Klot.

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
