## [Decision Letter · Decision Letter 0]

15 Jun 2023

PONE-D-23-06608ShinyLUTS - A Shiny web application for structured data management and analysis for patients with lower urinary tract symptoms (LUTS)PLOS ONE

Dear Dr. von Klot,

Thank you for submitting your manuscript to PLOS ONE. After careful consideration, we feel that it has merit but does not fully meet PLOS ONE’s publication criteria as it currently stands. Therefore, we invite you to submit a revised version of the manuscript that addresses the points raised during the review process.

ACADEMIC EDITOR:Please incorporate comments made by reviewers.

We look forward to receiving your revised manuscript.

Kind regards,

Ankit Gupta

Academic Editor

PLOS ONE

Journal Requirements:

2. Please amend your Data availability statement to provide details of where the data, code and application used and developed in this study can be found.

Additional Editor Comments (if provided):

Please incorporate comments made by reviewers.

Reviewers' comments:

Reviewer's Responses to Questions

**Comments to the Author**

1. Is the manuscript technically sound, and do the data support the conclusions?

Reviewer #1: Yes

Reviewer #2: Yes

Reviewer #3: Partly

2. Has the statistical analysis been performed appropriately and rigorously? 

Reviewer #1: N/A

Reviewer #2: I Don't Know

Reviewer #3: N/A

3. Have the authors made all data underlying the findings in their manuscript fully available?

Reviewer #1: Yes

Reviewer #2: Yes

Reviewer #3: Yes

4. Is the manuscript presented in an intelligible fashion and written in standard English?

Reviewer #1: Yes

Reviewer #2: Yes

Reviewer #3: Yes

5. Review Comments to the Author

Reviewer #1: Summary: Using the programming language R and the existing framework Shiny, the authors developed the web application ShinyLUTS to manage data of male patients with LUTS treated with Rezum therapy. The application consisted of 11 data tables. It appeared useful for scientific and clinical goals and could be used without technical knowledge of database systems.

Major comments:

1. The authors state that their work had two goals: (i) to improve the scientific management of data in a group of patients with LUTS and (ii) to improve the monitoring of individual patients with LUTS. The manuscript is however in fact only understandable for readers with knowledge of database structures and, in general, not for researchers and physicians in the field of urology. I would suggest to explore the possibilities to write the manuscript in a way that is more comprehensible for the latter group.

2. The results are described in a qualitative way and not quantitatively, which in a study like this hardly possible. This makes review questions 1-3 difficult to answer.

Minor comments:

1. It is unclear who physician TW (line 86) is. Dr. Winkler from ref. 5?

2. The authors report that 8118 parameter values were entered but not how many men were involved.

Reviewer #2: In the current study, the authors present the development and testing of a relational database model that enables researchers to enter highly complex and multidimensional data for patients with LUTS via a greatly simplified web interface into a modifiable database that can be used for daily clinical practice as well as large scale scientific evaluation of data. For this, they use the Shiny framework of the statistical programming language R.

The only question is about the patients? Why only patients with non-neurogenic male LUTS, including benign prostatic obstruction (BPO), were included in the first dataset?

Reviewer #3: The paper presents the development of web application using Shiny R package. The idea for structured data management is good. But, there are a few queries that need to be worked upon:

line 37- Citations must be provided for your claims.

The data description (schema) for patients must be mentioned in the paper.

Line 95-96 Why multiple format of data storage has been used. Authors must explain the purpose of using several packages in their work?

The GUI details must be rewritten. Clarity of sentences are missing.

line 121-22 -“The id column and the date column must have the same name in all tables.” why it is required? The names can always be aliased.

line 124- What do you mean by sub tables?

line 132-what are the pre-specified requirements?

The approach for creation of databases is unique but the paper must include the normalization aspects to handle inconsistencies. Also, the authors must specify how this work is worth executing? ie will there be any improvement to the data management aspects and services to the patients?

6. PLOS authors have the option to publish the peer review history of their article (what does this mean?). If published, this will include your full peer review and any attached files.

Reviewer #1: **Yes: **Jan Groen

Reviewer #2: No

Reviewer #3: **Yes: **Shweta Sinha

---

## [Author Response · Author response to Decision Letter 0]

18 Aug 2023

(please see the Response to Reviewers.docx)

We thank the Reviewers for the constructive remarks on our manuscript. We hope we have found the right balance between the demand for a ‘less technical, medical’ and a ‘more technically detailed’ manuscript. We hope the manuscript gives interesting new insight into an exciting topic. Please find our changes below:

Reviewer remarks are marked blue

Replies to reviewers are marked green

Changes/additions in the manuscript are marked red

References to the text without change are not marked

Reviewer #1: Summary: Using the programming language R and the existing framework Shiny, the authors developed the web application ShinyLUTS to manage data of male patients with LUTS treated with Rezum therapy. The application consisted of 11 data tables. It appeared useful for scientific and clinical goals and could be used without technical knowledge of database systems.

Major comments:

1. The authors state that their work had two goals: (i) to improve the scientific management of data in a group of patients with LUTS and (ii) to improve the monitoring of individual patients with LUTS. The manuscript is however in fact only understandable for readers with knowledge of database structures and, in general, not for researchers and physicians in the field of urology. I would suggest to explore the possibilities to write the manuscript in a way that is more comprehensible for the latter group.

Dear Reviewer, thank you for this very fitting remark. It addresses exactly the major challenge of our work: The main heading above the problem we are addressing could be the unification of two worlds: On the one hand the user side of a highly specialized physician, on the other hand the highly specialized field of activity of a database engineer. The incompatibility of these two worlds is also evident in the reviewers' comments: while some comments call for a more medically understandable text, others call for an even more technically detailed description of our work. We have tried to make the text interesting for both aspects. The introduction and the discussion are more user-oriented and focus on the problems of data acquisition from the physician's side. The material and methods part is more technical. In fact one reviewer even asked for more technical details. We added/made the following changes to make the text better understandable:

The paragraph was added (lines 38-41):

Urodynamic measurements generate vast amounts of data, are sometimes repeated and have a specific temporal relationship with therapeutic events or clinical outcomes, such as questionnaires. For clinicians it is sometimes difficult to maintain a reliable overview

The names of the main data tables was changed:

The names main and subtables were changed to “patient base table” and “clinical data table”.

We added the chapter “Use case” for better identification of a clinical/scientific application

We added an explanation and a new reference (lines 208 – 210): 

workflow with a Rmarkdown script for research purposes (RMarkdown is a markup language that allows to combine scientific manuscript writing and statistical calculations with the R programming language in one document [1]).

We corrected the sentence (line 251): “…Probably one of the major advantages of our application in scientific work was the possibility of centralized data entry by all participating physicians.

For better understandability we reworked the entire description of the application main interface (also requested by reviewer 3) (lines 123-):

Fig 1. General User Interface (GUI). The Figure shows the ShinyLUTS application interface. A) Browser window. B) Patient ID as a dropdown menu that allows manual input with autocompletion. C) A Summary of all data tables for the selected patient, defined by the patient ID in the dropdown menu (B): The summary shows the number of entries for each data table (n), the name of the data table (category), and the date of the last entry for each table (max_dat). This allowed for a quick overview of each patient. This aggregated view allows the user to quickly see how much data are available for the selected patient and when the last entry is dated. D) Dropdown menu for selecting the data table to be viewed or edited (Data Selection). E) Action buttons have various functions. ‘Enter new data’ will add a new empty row to the data table pre-selected in dropdown menu (D) below the data table (G), starting with the patient ID pre-selected in dropdown menu (B). Patient ID was not prefilled when the patient’s master table was selected. ‘Export all data / backup’ will create a dated backupfolder with a copy of the current database. The ‘SAVE’ and ‘QUIT’ buttons have corresponding functions. (F) Information box displaying a plausibility check. Because the underlying database is based on an Excel worksheet with the claim to be as free as possible, the usual restrictions on data consistency, normalization rules, or data types are not possible at the database level. Currently, our tool is limited to checking the patient ID of the master table and data table as part of the plausibility check. G) Selected data. The selected patient ID (B) and selected data table (D) determine the content of this table. In this view, data can be manipulated and new rows can be inserted via the ‘Enter new data’ action button. H) Graphical representation of data over time in relation to specific therapeutic interventions (Rezum™ in this case). The graphs show all numerical values of the selected data table with time/date references for the selected patient. This graphical representation was designed to monitor patient data over time in relation to a specific intervention.

We have added a short summary at the end of the discussion, emphasizing the clinical use of our new tool (lines 316):

We believe that our tool is worth implementing for various reasons: We have observed a significant improvement in the management of clinical data. In particular, patient monitoring has improved. For the first time, doctors have the opportunity to view all clinical data, such as therapeutic response, measurement or laboratory values when advising patients. We gained better work experience and increased clinical data availability. Our solution is scalable without increasing the complexity used by researchers and clinicians in the past, and was successful in the first scientific use case.

2. The results are described in a qualitative way and not quantitatively, which in a study like this hardly possible. This makes review questions 1-3 difficult to answer.

Dear Reviewer, thank you for this remark. It is true: This paper describes a methodology, not the results of a data analysis. The clinical results of REZUM therapy have been published in a different paper, which is also cited in the manuscript.

Minor comments:

1. It is unclear who physician TW (line 86) is. Dr. Winkler from ref. 5?

Dear Reviewer, thank you for this remark. Yes, MW will be referenced accordingly (line 94).

We added: (CAJK, TW **NEW CITATION:** [@winkler2023], MW). 

2. The authors report that 8118 parameter values were entered but not how many men were involved.

Dear Reviewer, thank you for this comment. The work presented has focused on the management of a large and overwhelmingly confusing amount of clinical data. While in a clinical study the number of subjects plays an important role for the significance of the results, for the presentation of our database tool the dimension of the data tables was the main focus (number of tables, number of data points). We have tried to focus on the data dimensions (important for a database) and not on the number of patients (important for a clinical trial) to avoid the false impression that we are talking about a statistical analysis. We will add the patient number by reviewer request.

The data table is also a work-in-progress with a constantly changing number of patients. 

We include a reference to the number of patients at the request of the reviewer with updated dimensions of the data structure as well as patient numbers at the first publication – we added (lines 111):

“….At the time of the first publication, the database consisted of 7 tables with 8,118 data points for 94 patients, 92 of which were to be evaluated in the publication [5]. As of today, the growing database organized by the ShinyLUTS tool includes 13 tables with 65,806 data points for a total of 243 patients….”

 

Reviewer #2: In the current study, the authors present the development and testing of a relational database model that enables researchers to enter highly complex and multidimensional data for patients with LUTS via a greatly simplified web interface into a modifiable database that can be used for daily clinical practice as well as large scale scientific evaluation of data. For this, they use the Shiny framework of the statistical programming language R.

The only question is about the patients? Why only patients with non-neurogenic male LUTS, including benign prostatic obstruction (BPO), were included in the first dataset?

Dear Reviewer, thank you for this question. At the time of the first publication, the database consisted of 7 tables with 8,118 data points for 94 patients, 92 of which were to be evaluated in the publication [5]. The involved researchers already had a project in mind that could then be published, which is why the group of included patients was initially restricted to those with LUTS/REZUM. The database however is rapidly growing and includes patients with other lines of therapy. The application appears to solve a decades-old data collection problem in our department, with likely expansion to other areas of urology. As of today, the growing database organized by the ShinyLUTS tool includes 13 tables with 65,806 data points for a total of 243 patients. Therefore, the reviewers commented that the tool may not be limited to patients with non-neurogenic male LUTS in the future is correct. 

We address this point at the end of the manuscript and modified the text accordingly (lines 330-): 

“….The valuable insight gained here in the course of the project about the requirements of physicians and researchers in the collection and analysis of clinical data is most likely not limited to patients with non-neurogenic male LUTS. As an outlook, we aim at a generalized form of our database solution for a variety of clinical projects….”

 

Reviewer #3: The paper presents the development of web application using Shiny R package. The idea for structured data management is good. But, there are a few queries that need to be worked upon:

line 37- Citations must be provided for your claims.

Dear Reviewer, thank you for this comment. We have added the following citations to the manuscript: 

1. Croll GJ, Butler RJ. Spreadsheets in clinical medicine - a public health warning. In 2006. 

2. Gordon KJ. Spreadsheet or database: Which makes more sense? Journal of Computing in Higher Education [Internet]. 1999 Mar;10(2):111–6. Available from: https://doi.org/10.1007/bf02948725

3. Wardle M, Sadler M. How to set up a clinical database. Practical Neurology [Internet]. 2015 Nov;16(1):70–4. Available from: https://doi.org/10.1136/practneurol-2015-001300

4. Mare I, Hazelhurst S, Kramer B, Klipin M. The process of installing REDCap, a web based database supporting biomedical research. Applied Clinical Informatics [Internet]. 2014;05(04):916–29. Available from: https://doi.org/10.4338/aci-2014-06-cr-0054

The data description (schema) for patients must be mentioned in the paper.

Dear Reviewer, thank you for this comment. Our concept is not subject to any rigid database structure and can, in principle, be modified freely by the user. For this reason, and because the text should not be kept too technical (in fact one reviewer asked for a less technical text), an illustration of the schema has been omitted so far. One of the key features of the application is that the user can constantly change the schema. However, for the rules controlled by the ShinyLUTS application, we have included the following schema into the GUI graphic together with a more thorough explanation of the data structure. 

In analogy to a typical database schema we added the following illustration:

(see image in the attached Response to Reviewers.docx)

Also, we added (lines 136):

Information box displaying a plausibility check. Because the underlying database is based on an Excel worksheet with the claim to be as free as possible, the usual schema with restrictions on data consistency, normalization rules, or data types are not possible at the database level. Currently, our tool is limited to checking the patient ID of the patient base table and clinical data table as part of the plausibility check.

Line 95-96 Why multiple format of data storage has been used. Authors must explain the purpose of using several packages in their work?

Dear Reviewer, thank you for this very good question. You are touching one of the major challenges with our work: On the one hand, the reader must understand the user side of a highly specialized physician in the field of urology, on the other hand the manuscript should be readable for a highly specialized database engineer. It was difficult for us to go into too much technical detail, since medical readers (and reviewers) demanded a less technical text. 

For many years, we have tried a wide variety of database solutions, some of which are also mentioned in the manuscript: Relational databases, non-relational databases, Microsoft Access, etc. All of these solutions offered a high level of data security, scalability, centralization, multi-user access, access rights management, data integrity, the use of foreign keys, Query options, etc.

Unfortunately, doctoral students and doctors and academic support staff could not be dissuaded from continuing to use Excel! In our department. Users indicated that they simply want to see the data, copy and move blocks, create new tables, create auxiliary tables, start and rename new columns, append intermediate calculations, they wanted variations in data types to be allowed and split the tables as they wished, etc., etc.

Dealing with data in this way is an absolute nightmare for an administrator of a professional database solution. To map this kind of freedom of Excel, for example, in SQL is impossible. The constant adjustment of the database schema alone was not reasonable and required permanent maintenance.

Indeed, SQLite was our first step into a less constrained database system. SQLite is lightweight, easily accessible, and tolerant of datatypes. In the end, however, all requirements could only be implemented if Excel was used. We mentioned this last step, i.e. the transition from SQLite to Excel, in the script, which is the reason why the package is mentioned.

A short, hopefully not to technical explanation of the packages is already provided in the methods section: 

“…..For the creation of the web based GUI, the packages: Shiny [@shiny2021], Shinymanager [@shinymanager2021] and DT [@dt2022] and for the connection to SQLite and Microsoft® Excel worksheet files, the packages RSQLite [@rsqlite2022] and openxlsx [@openxlsx2021] were used.”

The switch from SQL to .xlsx is explained here: 

“The first model for data management was a server based relational SQL database. In a completely different approach, we tested Microsoft® data tables in .xlsx- and .csv formatting. While SQL databases give great functionality with regard to data integrity, plausibility, the avoidance of redundant or inconsistent data and also facilitates normalization rules and constraints, we changed the data base backend to a simple spread sheet matrix located on a network drive which was maintained mainly by Microsoft® Excel. By doing so, clinicians and researchers were able to freely create tables, new fields and columns as desired.”

We specifically point out this ciritcal fact here:

“…the control of data integrity, relations between primary and external conclusions, 

and the plausibility of the data through the user's ability to access the data must be viewed critically. Our solution does not prevent inconsistencies in the data, but points them out.”

The GUI details must be rewritten. Clarity of sentences are missing.

Dear Reviewer, thank you for this comment

The GUI is described in three distinct positions in this manuscript. 

- First: the used package for the implementation of the GUI in the METHODS section

- Second: A brief description of the RESULTS section.

- Third: A description of the Figure describing the GUI

The description in the RESULTS section and the description of the GUI have been fused into one, better understandable, description of the Figure (which will appear below the Figure in the publication).

We have made the following changes by request of the reviewer:

A shorter and more concise description in the RESULTS section (lines 117-):

The final interface shows the patient master data with mandatory fields for the hospital-specific ID and date of birth. In a separate list, an aggregated overview of all clinical data tables and the date of the last entry in each table are shown. Each clinical data table can be selected individually for data manipulation. Lastly, a graphical representation of numerical data over time was implemented. Fig 1 shows the ShinyLUTS web interface in its final application form with a detailed description of functionality.

A detailed and clear description together with Fig. 1 was added (lines 123-):

Fig 1. General User Interface (GUI). The Figure shows the ShinyLUTS application interface. A) Browser window. B) Patient ID as a dropdown menu that allows manual input with autocompletion. C) A Summary of all data tables for the selected patient, defined by the patient ID in the dropdown menu (B): The summary shows the number of entries for each data table (n), the name of the data table (category), and the date of the last entry for each table (max_dat). This allowed for a quick overview of each patient. This aggregated view allows the user to quickly see how much data are available for the selected patient and when the last entry is dated. D) Dropdown menu for selecting the data table to be viewed or edited (Data Selection). E) Action buttons have various functions. ‘Enter new data’ will add a new empty row to the data table pre-selected in dropdown menu (D) below the data table (G), starting with the patient ID pre-selected in dropdown menu (B). Patient ID was not prefilled when the patient’s master table was selected. ‘Export all data / backup’ will create a dated backupfolder with a copy of the current database. The ‘SAVE’ and ‘QUIT’ buttons have corresponding functions. (F) Information box displaying a plausibility check. Because the underlying database is based on an Excel worksheet with the claim to be as free as possible, the usual restrictions on data consistency, normalization rules, or data types are not possible at the database level. Currently, our tool is limited to checking the patient ID of the master table and data table as part of the plausibility check. G) Selected data. The selected patient ID (B) and selected data table (D) determine the content of this table. In this view, data can be manipulated and new rows can be inserted via the ‘Enter new data’ action button. H) Graphical representation of data over time in relation to specific therapeutic interventions (Rezum™ in this case). The graphs show all numerical values of the selected data table with time/date references for the selected patient. This graphical representation was designed to monitor patient data over time in relation to a specific intervention.

line 121-22 -“The id column and the date column must have the same name in all tables.” why it is required? The names can always be aliased.

Dear Reviewer, thank you for this comment. 

Proper column names are necessary for the data tables to be automatically integrated into the ShinyLUTS tool. One of the major goals of our tool was to give most control the researcher without the need for an administrator to intervene. Theoretically, different column names with aliases would be possible; however, this would mean that these aliases have to be defined. Currently, only strict rules apply to the name of the patients’ ID and the date within a tuple. There should be no effort for the researcher to adhere to these minimal conventions.

We added (line 159): 

We adhered from strict column name convention for the patients’ ID and the date column to avoid the definition of aliases.

line 124- What do you mean by sub tables?

Dear Reviewer, thank you for this very good question. As mentioned in the previous comments, our approach does not follow a strict scheme, as expected for an SQL database.

We divided the data logically into

1. A main table (in other words: ‘patient base table’ or ‘patient master table’ and 

2. None or several so called sub tables (in other words: ‘clinical data tables’).

Each patient has unchangeable data, such as date of birth, last name, first name, and patient ID. Each patient must exist in the main table with the according data and patients’ ID.

In addition to the main table, several tables of clinical data are available. These tables were referred to as sub-tables. Since all data in medicine have a time dimension (death, medication, symptoms, treatment, imaging, etc.), all sub-tables must start with the patients’ ID (corresponding to the main table) and a date (at which the laboratory value, medication, symptoms..etc occurred). 

The ShinyLUTS tool provides access to the main table and implements and displays all sub-tables (that have patients’ ID and dates as the first two columns). Researchers have total freedom of including helper-tables, notes, diagrams, graphics as separate tables and can edit and view them within Excel, they will, however not be implemented in ShinyLUTS if they do not have the columns patients’ ID and date. 

For better understanding, we changed the names ‘main table’ to ‘patient base table’ and the name ‘subtable’ to ‘clinical data table’ by Reviewer request in the entire manuscript and the images.

line 132-what are the pre-specified requirements?

Dear Reviewer, thank you for the comment. The sentence is in fact not correct. It was changed from: 

The pre-specified requirements for the integration of spread-sheet functionality were therefore fully met.

To (line 169):

The predefined requirement to ensure spreadsheet functionality was thus fully met.

The approach for creation of databases is unique but the paper must include the normalization aspects to handle inconsistencies.

Dear Reviewer, thank you for the comment. We knew that normalization rules help ensure that data is organized efficiently and without redundancy and it helps to prevent data duplication and inconsistencies. Normalization rules 1NF, 2NF, and 3NF… can be implemented using fixed database schemas. Table's name, columns, constraints, data types, relationships, and other properties can be applied in a variety of database management systems. However, by giving the researcher the demanded full control of the data within the Excel environment (including for example copy paste operations of large blocks of data), normalizsation rules cannot be enforced on the side of the database backend and they cannot be controlled by the ShinyLUTS frontend side because direct access to the Excel database is always possible. To address this problem, we implemented data validation in ShinyLUTS. This means that, for example, the patient ID in the data tables must appear in the main table. If this is not the case, data entry is not prevented, but the inconsistency of the data is displayed as a warning in the message field within ShinyLUTS. An extension of this verification would be the control and warning of duplicates or warnings in case of false data types. The concept of our solution requires that the data input is not restricted, but violations of the normalization rules are indicated as warning message. 

This concept is explained in the manuscript:

“… While SQL databases give great functionality with regard to data integrity, plausibility, the avoidance of redundant or inconsistent data and also facilitates normalization rules and constraints, we changed the data base backend to a simple spread sheet matrix located on a network drive which was maintained mainly by Microsoft Excel. By doing so, clinicians and researchers were able to freely create tables, new fields and columns as desired. All tables and fields

… Following this approach, the restraints of an SQL database could no longer be implemented. However, we created an adaptable plausibility check on the level of the application as info box inside the ShinyLUTS GUI. At the current development state, the application will check (but not prevent) for inconsistencies between patient ids in the patient base table and in the clinical data table.”

Updated text (line 136):

“…(F) Information box displaying a plausibility check. Because the underlying database is based on an Excel worksheet with the claim to be as free as possible, the usual schema with restrictions on data consistency, normalization rules, or data types are not possible at the database level. Currently, our tool is limited to checking the patient ID of the patient base table and clinical data table as part of the plausibility check.”

 Also, the authors must specify how this work is worth executing? ie will there be any improvement to the data management aspects and services to the patients?

Thank you for your comments. In all honesty, the current solution is a huge improvement for doctors, researchers, and patients. For more than a decade, we have been looking for an optimization of our patient data for better control of therapy and diagnostics, as well as for a better possibility of scientific evaluation. The current solution, which we would like to share in this publication, is the first one that seems to work as intended for the first time in years. With data within a spreadsheet solution, smaller projects were possible in the past, but longer-term projects or increasingly complex amounts of data with constantly changing statistical clinical questions simply overwhelmed researchers with this form of data entry method. 

With the current solution, researchers and doctors are able to distribute the data across multiple tables without losing track. In addition, errors in data consistency are reported to the user immediately and are clearly visible. We see that, as more data and new projects are added, the complexity does not increase for the user, which is a significant improvement.

We have added the section ‘Use case’ to better identify usability within a research project (line 205):

Use case 

“As part of an initial retrospective research project,…..”

A great scientific advantage is the R environment, in which the entire project was designed. The scientific evaluation mentioned in this work was carried out using an R-Markdown script with the ShinyLUTS Excel spreadsheet as the data source. This allowed researchers to create a scientific paper for retrospective data analysis simultaneously, as data were entered. Statistical tests, graphs, tables, patient characteristics, references, and the entire manuscript were created and repeatedly compiled via Pandoc as the amount of data increased. This unique, reproducible form of analysis was made significantly easier using ShinyLUTS.

We altered/added text (without making it too technical (by request of Reviewer 1)):

“…also into a workflow with a Rmarkdown script for research purposes (RMarkdown is a markup language that allows to combine scientific manuscript writing and statistical calculations with the R programming language in one document [@rmarkdown]). Data were selected, filtered and aggregated according to the research question. The researchers were able to observe the progress of the data….”

The application is also focused on the data for patients with LUTS, that is, a graphical representation showing the clinical parameters in relation to certain therapies. For the first time, doctors have the opportunity to view all clinical data, therapeutic successes, measurement data, and laboratory values when advising patients. Even data that are not in the patient's file but are recorded as part of a research project, for example, are visible. 

We explained this benefit for the patient and the treating clinician in the figure description of Fig.1:

H) Graphical representation of data over time in relation to specific therapeutic interventions (Rezum; in this case). The graphs show all numerical values of the selected data table with time/date references for the selected patient. This graphical representation was designed to monitor patient data over time in relation to a specific intervention.

The data were immediately filterable, sortable, and presented in an aggregated form that was never seen before in our normal patient files. New data can be entered seamlessly via the tool during doctor-patient consultation, and the data are automatically entered into the corresponding Excel spreadsheets, which is a great advantage for us and the patient. Because the data are stored centrally, doctors can access it from all workstations in our outpatient clinic, so doctors who do not have the patient file at the moment can also contribute, which is another advantage. A great scientific advantage is the R environment, in which the entire project was designed. The scientific evaluation mentioned in this work was carried out using an R-Markdown script with the ShinyLUTS Excel spreadsheet as the data source. This allowed researchers to create a scientific paper for retrospective data analysis simultaneously, as data were entered. Statistical tests, graphs, tables, patient characteristics, references, and the entire manuscript were created and repeatedly compiled via Pandoc as the amount of data increased. This unique, reproducible form of analysis was made significantly easier using ShinyLUTS.

By Reviewer request, we have added a summary on why the work is worth executing (line 316):

We believe that our tool is worth implementing for various reasons: We have observed a significant improvement in the management of clinical data. In particular, patient monitoring has improved. For the first time, doctors have the opportunity to view all clinical data, such as therapeutic response, measurement or laboratory values when advising patients. We gained better work experience and increased clinical data availability. Our solution is scalable without increasing the complexity used by researchers and clinicians in the past, and was successful in the first scientific use case.

 

---

## [Decision Letter · Decision Letter 1]

13 Sep 2023

ShinyLUTS - A Shiny web application for structured data management and analysis for patients with lower urinary tract symptoms (LUTS)

PONE-D-23-06608R1

Dear Dr. von Klot,

We’re pleased to inform you that your manuscript has been judged scientifically suitable for publication and will be formally accepted for publication once it meets all outstanding technical requirements.

Kind regards,

Ankit Gupta

Academic Editor

PLOS ONE

Additional Editor Comments (optional):

Reviewers' comments:

Reviewer's Responses to Questions

**Comments to the Author**

1. If the authors have adequately addressed your comments raised in a previous round of review and you feel that this manuscript is now acceptable for publication, you may indicate that here to bypass the “Comments to the Author” section, enter your conflict of interest statement in the “Confidential to Editor” section, and submit your "Accept" recommendation.

Reviewer #1: All comments have been addressed

Reviewer #3: All comments have been addressed

2. Is the manuscript technically sound, and do the data support the conclusions?

Reviewer #1: Yes

Reviewer #3: Partly

3. Has the statistical analysis been performed appropriately and rigorously? 

Reviewer #1: N/A

Reviewer #3: No

4. Have the authors made all data underlying the findings in their manuscript fully available?

Reviewer #1: Yes

Reviewer #3: Yes

5. Is the manuscript presented in an intelligible fashion and written in standard English?

Reviewer #1: Yes

Reviewer #3: Yes

6. Review Comments to the Author

Reviewer #1: 1. The authors comprehensively replied to the reviewers’ remarks and questions. This greatly improved the manuscript.

2. The authors show the possibilities of database systems and it is to be expected that their methods can relatively simple be transferred to other fields of medicine.

Reviewer #3: All the raised concern in the first round has been resolved. Even though the information covered in the paper is adequate , the authors may extend the statistical analysis to cover more aspects of data.

7. PLOS authors have the option to publish the peer review history of their article (what does this mean?). If published, this will include your full peer review and any attached files.

Reviewer #1: **Yes: **Jan Groen

Reviewer #3: **Yes: **shweta sinha

---

## [Editor Report · Acceptance letter]

19 Sep 2023

PONE-D-23-06608R1 

ShinyLUTS - A Shiny web application for structured data management and analysis for patients with lower urinary tract symptoms (LUTS) 

Dear Dr. von Klot:

I'm pleased to inform you that your manuscript has been deemed suitable for publication in PLOS ONE. Congratulations! Your manuscript is now with our production department. 

Kind regards, 

on behalf of

Dr. Ankit Gupta 

Academic Editor

PLOS ONE